# Detecting Multimodal Situations with Insufficient Context and Abstaining from Baseless Predictions

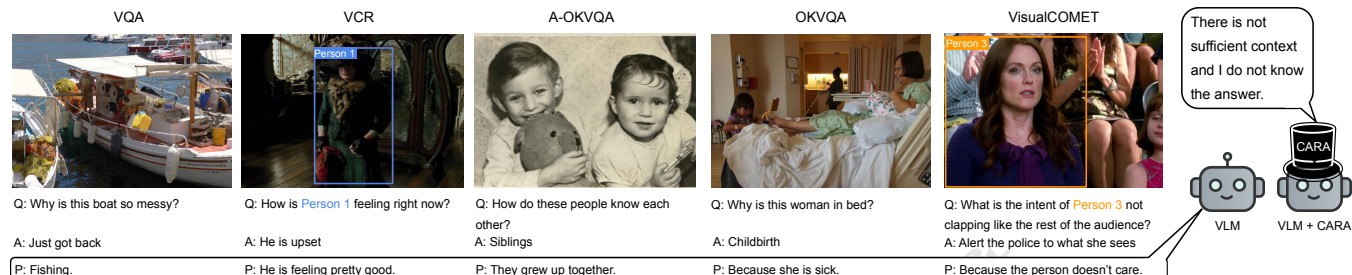

**Figure 1: Examples of samples with insufficient context to answer the question across several representative Vision Language Understanding (VLU) benchmarks. "Q" represents the question, "A" stands for the answer, and "P" denotes a typical Vision Language Model (VLM; here BLIP-2) prediction. We find that samples with insufficient context are common across several VLU benchmarks, causing VLMs to hallucinate predictions. Using (wearing) our proposed CARA (hat), VLMs are able to abstain from responding instead of making baseless predictions in such cases.**

## ABSTRACT

Despite the widespread adoption of Vision-Language Understanding (VLU) benchmarks such as VQA v2, OKVQA, A-OKVQA, GQA, VCR, SWAG, and VisualCOMET, our analysis reveals a pervasive issue affecting their integrity: these benchmarks contain samples where answers rely on assumptions unsupported by the provided context. Training models on such data fosters biased learning and hallucinations as models tend to make similar unwarranted assumptions. To address this issue, we collect contextual data for each sample whenever available and train a context selection module to facilitate evidence-based model predictions. Strong improvements across multiple benchmarks demonstrate the effectiveness of our approach. Further, we develop a general-purpose Context-AwaRe Abstention (CARA) detector to identify samples lacking sufficient context and enhance model accuracy by abstaining from responding if the required context is absent. CARA exhibits generalization to new benchmarks it wasn't trained on, underscoring its utility for future VLU benchmarks in detecting or cleaning samples with inadequate context. Finally, we curate a Context Ambiguity and Sufficiency Evaluation (CASE) set to benchmark the performance of insufficient context detectors. Overall, our work represents a significant advancement in ensuring that vision-language models generate trustworthy and evidence-based outputs in complex real-world scenarios.

**ACM Reference Format:**
. 2024. Detecting Multimodal Situations with Insufficient Context and Abstaining from Baseless Predictions. In *Proceedings of (MM '24)*. ACM, New York, NY, USA, 10 pages. https://doi.org/XXXXXXX.XXXXXXX

## 1 INTRODUCTION

A number of Vision Language Understanding (VLU) benchmarks have been proposed to evaluate the capability of models to interpret complex multimodal scenarios and events [1, 28, 30, 38, 53, 55]. However, these benchmarks often include samples with insufficient event-specific context to answer the given questions. For instance, in the first example of Figure 1, it is impossible to answer why the boat is messy without knowing what had happened before. Similarly, in the second example of Figure 1, knowledge of [Person 1]'s prior interaction is required to determine how the person feels. Answering the questions for these examples requires more contextual information about the events depicted in the images than is available from the image alone.

Our analysis reveals that this issue of insufficient event-specific context is pervasive in many VLU datasets. Figure 1 illustrates examples from some representative benchmarks – VQA v2 [1], Visual Commonsense Reasoning (VCR) [54], OKVQA [28], A-OKVQA [38], and VisualCOMET [30]. The lack of sufficient and specific context in the provided samples forces models trained on such data to guess possible answers, leading to models that confidently predict answers without evidential support. Models that tend to hallucinate assumptions in this way undermine their trustworthiness and limit their real-world applicability in settings where accuracy is critical e.g., assistive technologies for the visually impaired [11, 31, 58], autonomous vehicles and robotics [12, 15], healthcare applications [25] or security and surveillance [45].

Our findings of the ubiquity of this problem lead us to two critical questions: 1) If the context can be retrieved, *e.g.*, we can obtain the corresponding video as context when the sample has an image from that video, how to identify the most necessary context and

effectively incorporate it into models? 2) If there is no available context, *e.g.*, the sample has an in-the-wild image, can we develop a generalizable method to identify samples with insufficient context and abstain from making baseless predictions?

Regarding the first question, numerous methods [19, 40] have been proposed to enhance image-text understanding with external knowledge. Yet, these approaches fail to address the absence of *event-specific* context, which is not available in external sources. The challenges presented in Figure 1, for example, cannot be overcome simply through the application of general knowledge, as they require insights directly related to the depicted events.

As for the second question, no prior work has focused on abstaining from speculative responses by identifying insufficient event-specific context across existing VLU benchmarks. Existing works refrain from answering either due to low model confidence [50] or due to out-of-distribution samples [6]. Consequently, they would still make unfounded predictions for samples with high model confidence or in-domain samples but insufficient context.

We address these limitations by 1) Collecting contextual data where available (VCR, SWAG, and VisualCOMET) and building a novel model-agnostic plug-and-play context selection module to incorporate context into model prediction (see Figure 2); 2) Reusing the aforementioned module to collect pseudo-labels to train a Context-AwaRe Abstention (CARA) module, capable of identifying samples with insufficient context. Both our context selection module and CARA are model- and task-agnostic.

Our experiments demonstrate that our context selection module consistently improves performance across models and tasks. In the process, we also investigate several important questions: 1) In which modality (visual, textual, or both) does context benefit the most? 2) How long is context useful before it becomes noise? Further, we show that CARA boosts state-of-the-art model performance in full-shot and zero-shot settings by reducing inaccurate predictions through abstention from baseless or hallucinated responses. CARA, trained on one benchmark, can effortlessly generalizes to other benchmarks as well. This provides evidence that CARA could be useful for future benchmarks as well without any re-training. It could even be used to clean future benchmarks of samples with insufficient context. Moreover, as CARA prevents models from making predictions that are not grounded in contextual evidence, we believe it will significantly improve model trustworthiness.

Lastly, to evaluate CARA's quality and benchmark its performance, we also curate an evaluation set manually annotated with the labels – sufficient or insufficient context. This data is valuable for the future development of insufficient context detectors.

In summary, our work makes several key contributions:

- **Highlighting a Systemic Issue**: We identify a pervasive problem in common VLU benchmarks, i.e., the presence of samples with insufficient context. This issue has been largely overlooked in prior studies, despite its impact on the performance and reliability of VLU models. We conduct an extensive analysis across benchmarks to reveal the extent of this problem.
- **Incoporating Context Effectively**: We address the issue of insufficient event-specific context in VCR, SWAG, and VisualCOMET benchmarks by introducing a novel context

selection method. This enhances model performance by accurately identifying and integrating relevant context into task resolution.
- **Multimodal Abstention Detector**: We develop CARA, a method for abstaining on samples lacking necessary context, and demonstrate its generalization across new benchmarks.
- **Data Contribution**: We collect contextual data for VCR, SWAG, and VisualCOMET, which is valuable for further exploration of context-aware model prediction. Moreover, we create a Context Ambiguity and Sufficiency Evaluation (CASE) set for insufficient context detection.

## 2 RELATED WORK

*2.0.1 Unanswerable Visual Questions.* The challenge of determining the answerability of visual questions has been explored primarily from two main directions: 1) relevance of the question or 2) quality of the image. The former direction focuses on creating datasets and methods that test models' ability to flag irrelevant questions [17, 22, 27, 33, 44] or questions inquiring about objects absent in the image [23, 26, 51]. On the other hand, the latter direction requires models to flag unanswerable samples due to low image quality [3, 11]. Both directions overlook the nuanced complexity of unanswerability in the case of insufficient context for high-quality images paired with relevant questions. It is this gap that our work aims to bridge.

*2.0.2 VLU with External Resources.* When information in the image is insufficient to answer the question [28, 36, 38], several methods have been proposed to augment the provided information with external knowledge from Wikipedia [24], the internet [14], and knowledge graphs [19, 40]. Our approach is similar in retrieving extra information to complement the provided visual information. However, we retrieve contextual information directly related to events and entities depicted in the image, while prior approaches search for general factual [2, 48] or commonsense knowledge [37, 41]. The contextual information we seek, for example, the reason for [Person 1]'s injury in Figure 2, is unavailable in those external sources. Limited works have explored specific sample-related contextual information. Naik et al. [29] utilize image source metadata while Biten et al. [4], Tran et al. [46] leverage paired news article. Both cases bypass context retrieval by exploiting image metadata as is, unlike our work. Furthermore, they do not focus on integrating temporal or event-specific context, which is crucial for reasoning in semantically complex VLU tasks.

*2.0.3 Abstention in Multimodal Systems.* Abstaining from responding instead of making incorrect predictions was originally explored in the unimodal language domain to address out-of-distribution or adversarial inputs [5, 7, 9, 16, 18, 47]. In the multimodal domain, recent works have been proposed that abstain similarly in the case of out-of-distribution samples [6] or low model confidence [50]. In contrast, our proposed approach avoids making predictions when sufficient context to answer the question is unavailable. Unlike prior works, our abstention mechanism works from a data-centric view and applies to new benchmarks without any re-training.

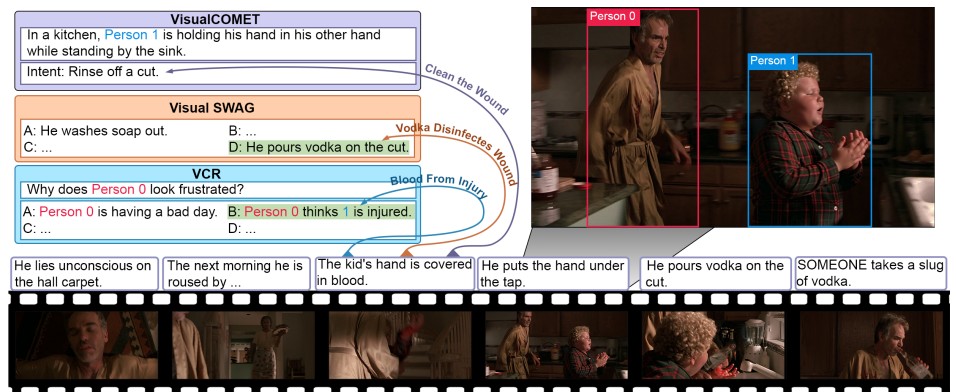

**Figure 2: Illustration of how we obtain contextual data for VCR, Visual SWAG, and VisualCOMET. The video from which the image sample is sourced is identified to obtain temporal context in the form of frames and captions around the image sample in question. The context provides the necessary evidence required to answer these highly semantic questions.**

## 3 PROBLEM SPACE

We investigate several benchmarks to study the problem of insufficient context in VLU domain: VQA v2, OKVQA, A-OKVQA, GQA, VCR, VisualCOMET, and Visual SWAG. These datasets cover a range of VLU tasks, such as visual question answering, image-based text generation, and image-text matching. Notably, SWAG is a text-only entailed event inference dataset. To facilitate the study on multimodal event entailment inference, we replace the text premise in SWAG with the corresponding image frame retrieved from the samples' source video. We call this multimodal dataset, Visual SWAG, where given an image premise, the required task is to infer the entailed event in textual form.

For the datasets with contextual data available – VCR, Visual-COMET, and Visual SWAG – we first collect that contextual data, as in Section 4. Then, we utilize it to facilitate evidence-based VLM prediction via a context selection module, as in Section 5.1. Further, we leverage a combination of vanilla VLM and VLM trained with context to pseudo-label samples with insufficient context. The pseudo labels are then used to train an insufficient context detector, CARA. We demonstrate that CARA generalizes to VQA v2, OKVQA, A-OKVQA, and GQA without having ever been trained on them.

The input information for the above reasoning benchmarks can be generally denoted as $x = (x_T, x_I)$ where $x_T$ is the textual input and $x_I$ is the image input. In our first study, we explore whether adding another input Context, $x = (x_T, x_I, C)$, can help and explore how to obtain the most beneficial context. For our second study, we develop functions to detect samples with insufficient context in the input, $x = (x_T, x_I)$, and abstain from making baseless predictions.

## 4 CONTEXTUAL DATA COLLECTION

We begin by collecting contextual data for the three VLU benchmarks described above. These benchmarks evaluate models' understanding of events using images sourced from existing video datasets. To ensure comprehensive context is provided for each sample across the tasks, we collected multimodal contextual data, including preceding and subsequent visual frames along with paired text scripts. We first discuss how we retrieved the source video data

and then how context was retrieved and filtered. Finally, we present statistics about our assembled dataset.

### 4.1 Data Fetching

The images from VCR, VisualCOMET, and Visual SWAG are derived from video sources like LSMDC [35], ActivityNet[10], or YouTube. These video datasets consist of sequences of video clips, where each clip is paired with a sentence describing the event in the clip. Since annotations in VCR, VisualCOMET, and Visual SWAG include specific frame IDs and clip IDs for most samples, we can locate and retrieve the source clip of the corresponding sample. We removed all samples for which we could not find the corresponding source clip or paired video scripts.

### 4.2 Context Retrieval and Filtering

*4.2.1 Context Retrieval.* We retrieve the clips before and after the corresponding source clip as visual context. These video clips are also paired with video scripts. We retrieve these scripts as text context for data points. However, using video frames as visual context can be highly redundant due to their repetitive nature (i.e. adjacent frames are generally very similar), thus we find the most descriptive frame from each of these clips by finding the best match with the script using a pre-trained CLIP [32] model. More formally, we denote context from preceding clips with negative indices ($c_{-3}, c_{-2}...$), while context from succeeding clips has positive indices ($c_1, c_2...$), where each $c_i$ consists of both vision and language contexts.

*4.2.2 Context Filtering.* Given that a substantial portion of our datasets comprises temporal questions, specifically those inquiring about states before and after, we take precautions to avoid inadvertently providing context that may disclose the answer to the model. We achieve this by identifying such cases using keywords and then filtering out contexts that could potentially lead to cheating. For instance, samples featuring questions about the past will be devoid of negatively indexed contexts.

### 4.3 Data Statistics

*4.3.1 Training Data.* Our training dataset split includes 41,008 image-text pairs from the train split of Visual SWAG, with an additional 94,404 distinct image-text pairs as multimodal context;

119,994 image-question pairs from the train split of VCR with 79,913 distinct image-text context pairs; and 190,457 image-text pairs (from 63,499 unique situations) from the train split of VisualCOMET, with 79,109 distinct image-text context pairs.

*4.3.2 Evaluation Data.* Our validation set includes 10,645 image-text pairs from the train split of Visual SWAG with 57,880 distinct image-text pairs as multimodal context; 15,092 image-question pairs from the train split of VCR with 10,014 distinct image-text context pairs; and 23,930 image-text pairs (from 7,978 unique situations) from the train split of VisualCOMET, with 9,933 distinct image-text context pairs.

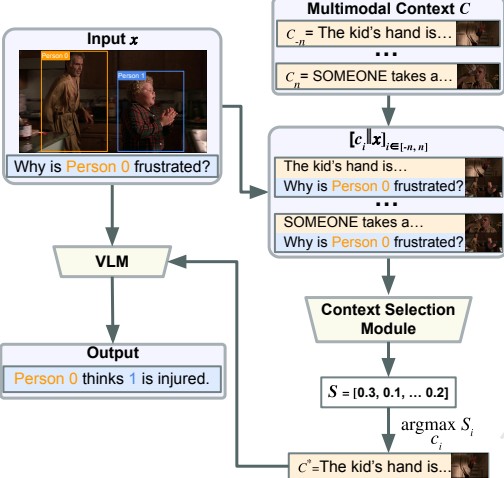

**Figure 3: A high-level demonstration of the probabilistic context selection method. For the VLM's input, in addition to the question and image, a context sentence selected by the Context Selection Module is appended to the original input.**

## 5 METHOD

Using our collected contextual data, we first develop a model-agnostic smart context selection module to add relevant context to samples to improve the model's understanding of the sample and, hence, its performance. We then create a multimodal abstention model to identify samples lacking sufficient event-specific context and prevent models from making baseless predictions on such samples.

### 5.1 Context Selection Module

Consider a Vision-Language Model (VLM) $M$, with image input $x_I$ and text input $x_T$. The most straightforward way to incorporate context, $C = [c_i]_{i \in [-n,n]}$, is to append it to the model input. That is, $y_{pred} = M(x, C)$, where $x = (x_I, x_T)$. This results in a brute-force context injection approach, which is both heavily computationally expensive and potentially noisy. Instead, we aim to build a method to intelligently select the most relevant context according to the given target image and text premises.

We thus propose a "probabilistic context selection" method (see Figure 3). This end-to-end, model-agnostic technique aims to streamline the selection of event-specific context. Our method features a

context selection module $M_c$ designed to identify the most relevant context $c^*$ for the given input $x$. As a result, the model's output is given by $y_{pred}^* = M(x, c^*)$. The core idea behind this approach is that it can dynamically select the context that is most aligned with the input. This allows it to integrate only the most relevant context into the downstream reasoning process while filtering out noisy context. We demonstrate that this significantly improves the model's ability to handle complex reasoning tasks requiring contextual information.

Specifically, for a given input $x$ and a set of context $c_i$, the selection module $M_c$ computes a score vector $S = [s_i]_{i \in [-n,n]} = M_c(x, c_i)$. Each score $s_i$ within this vector denotes the relevance of $c_i$ to $x$. The $c_i$ with the highest $s_i$ is chosen as the selected context for inference. During training, each $s_i$ is used to (softly) select the $c_i$ as the context in the VLM, $M$. This encourages the context selection module $M_c$ to assign a low weight to context $c_i$, which leads to a high loss in $M$ and vice versa. Thus, $M_c$ is trained to assign a high weight to the most relevant context. The resulting loss function is:

$$\mathcal{L} = \sum_{i=-n}^{n} s_i \cdot l(M(x, c_i), y) \tag{1}$$

where, $l$ represents the cross-entropy loss.

This probabilistic sampling procedure, where $c_i$ is sampled using $s_i$, is differentiable end-to-end. We illustrate how the context selection module interacts with the backbone VLM in Figure 3.

For a given input $x$ and a specific context $c_i$, we append the context with the input to create $x_i = [x \| c_i]$. More specifically, text context $c_{i_T}$ is appended to text input $x_T$ and image context $c_{i_I}$ with image input $x_I$, creating $x_{i_T} = [x_T \| c_{i_T}]$ and $x_{i_I} = [x_I \| c_{i_I}]$ respectively. $x_{i_T}$ and $x_{i_I}$ are then processed by $M$ as it would normally process $x_T$ and $x_I$.

### 5.2 Multimodal Abstention Detector

The above section assumes additional context is available to be recovered through retrieval. However, in many real-world scenarios, additional context may not be available. Thus, we propose a generalized multimodal abstention detector that aims to identify if a sample is unanswerable due to a lack of context thereby preventing baseless predictions.

Developing a mechanism to detect samples with insufficient event-specific context is an extremely challenging problem because the model must first hypothesize what the sufficient context would be to answer the question before determining if that context is lacking. In this work, we present a straightforward yet effective solution to address this issue. We leverage our previously trained model with context and compare its response with a vanilla model trained without context to pseudo-label if the sample contains or lacks sufficient context. Our key insight is that if a sample already has sufficient context, the model's response should remain relatively consistent when additional context is added. Conversely, if the sample lacks sufficient context, the model's response should improve on adding additional context. The pseudo labels are then used to train our insufficient context detector. We illustrate this process in Figure 4 and detail it below.

*5.2.1 Confidence-Driven Pseudo-Labelling.* We train two models: a Context-VLM (C-VLM), which incorporates context into its decision-making process (as detailed in Section 5.1, and a vanilla VLM, which operates without context. We compare the responses from both models to pseudo-label samples as follows:

- **Positive**: Instances correctly answered by the C-VLM model with high confidence above a designated threshold, $\gamma$, but incorrectly answered by the VLM with low confidence below a designated threshold $\mu$. The significant difference in accuracy and confidence suggests that these instances previously lacked sufficient context for unambiguous understanding.
- **Negative**: Instances correctly recognized by both models with confidence above threshold $\gamma$, implying that context does not play a critical role in their identification.
- **Excluded**: Instances not fitting into the above categories are excluded. The impact of context on these instances is uncertain, and including them could introduce noise into the model training process.

When pseudo-labeling the training set, we divide the dataset into two equal parts. Each part is labeled based on the inference results obtained from the two models trained on the other half. This strategy ensures a robust pseudo-labeling process that mitigates overfitting risks and data leakage from the validation set.

*5.2.2 Training.* After pseudo-labeling the data points, we train CARA, a 24-layer cross-modal attention module as our insufficient context detector via cross-entropy loss. The input to CARA is a sample image and the corresponding question or statement, and the output is a binary label denoting whether the sample lacks sufficient context or not.

*5.2.3 Inferencing.* When inferencing using CARA, we predict if a data point lacks sufficient context based on whether CARA's prediction score exceeds a dataset-specific threshold $\theta$.

Note that during both training and inference, the detector operates *without access to context*. Our goal is to develop a generalized detector that maintains high performance and generalizability across datasets, regardless of whether context is available.

## 6 EXPERIMENTS

In this section, we present experimental results and analysis to illustrate the effectiveness of our context selection methodology and CARA. We first present implementation details, followed by context selection results and abstention detection results.

### 6.1 Implementation Details

*6.1.1 Base Models.* We demonstrate the efficacy of our approach on two different classes of VLMs: discriminative (VL-BERT[42]) and generative (BLIP [21], BLIP2 [20], MCAN[52], MiniGPT-4 [57], OFA [49], PNP[43], Prophet[39], and PromptCap[13] ). Generalization across models shows that our approach is model-agnostic. We adhered strictly to the implementation described in the original papers and repositories of all models.

*6.1.2 Training.* Fine-tuning of VL-BERT, BLIP, and OFA is done with 2 NVIDIA-RTX 24 GB GPUs with batch size 32, BLIP2 and MiniGPT-4 are trained with 2 NVIDIA A100 GPUS with the same

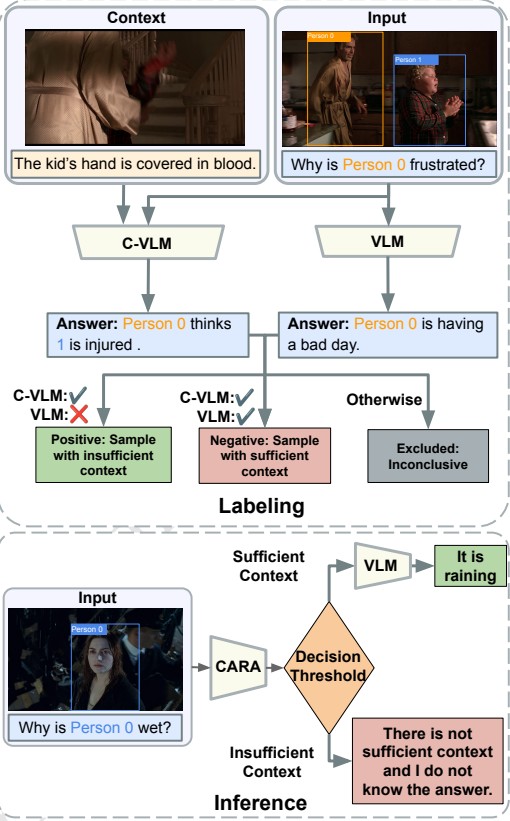

**Figure 4: Top: We use models with/without context to pseudo-label whether instances need context. The labeled data is then used to train CARA. Bottom: CARA decides whether to abstain based on whether the input contains sufficient context.**

batch size. The initial learning rates for VL-BERT, BLIP, BLIP2, MiniGPT-4, and OFA are 7e-5, 1e-5, 2e-5, 3e-5, and 3e-5, respectively. VL-BERT is trained for 20 epochs, and BLIP is for 10 epochs, while BLIP2 and MiniGPT-4 are trained for 5 epochs. For OFA, we follow the original implementation and train a total of 40K steps. Training of the models takes ~48 hours. The abstention detector is trained for 10 epochs with a learning rate of 7e-5.

*6.1.3 Context Selection.* BLIP, BLIP2, MiniGPT-4, and OFA lack native RoI functions as in VL-BERT. Thus, to process datasets requiring RoI handling such as VCR and VisualCOMET we adopt Merlot's approach [56] of drawing colored highlights around referenced entities in pixel space, as shown in Figure 2. In our experiments, we employ a Sentence-BERT [34] as the text encoder for $M_C$, and a ViT [8] as the vision encoder. We fuse the global embeddings from those two encoders' output via concatenation and apply an MLP with sigmoid to map the fused feature into a score ranging from 0 to 1.

*6.1.4 Abstention Detector.* We use a 24-layer cross-modal attention model as the multimodal abstention detector, following [42] to initialize and train it on datasets labeled with the pseudo-labeling method described in Section 5.2. However, this results in an unbalanced training set with significantly more negative data points. To

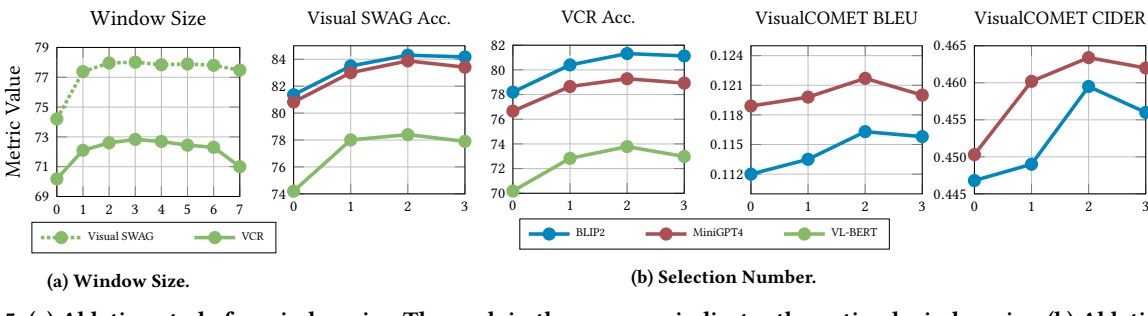

(a) Window Size.    (b) Selection Number.

Figure 5: (a) Ablation study for window size. The peak in the accuracy indicates the optimal window size. (b) Ablation study for selection number. We allow the VLM to observe multiple contexts within the window size. The peak in the curve indicates selecting 2 out of 3 using the probabilistic selection method results in the highest evaluation result.

address this, we apply loss weighting during training. Our experiments show increasing the weight of positive data points in the loss by six results in the highest evaluation performance.

The CARA tailored for the Visual SWAG and VCR tasks are trained on their respective datasets. However, since VisualCOMET is a generative dataset without a binary correctness measure, we utilize the CARA trained on VCR for it instead. In practice, we utilize heuristics rules [1] integrating the confidence scores of the VLM and CARA's prediction to obtain the best model performance over downstream tasks.

When conducting confidence-driven pseudo-labeling, the hyperparameters for filtering thresholds, $\gamma$, is 0.7 and $\mu$ is 0.5.

## 6.2 Context Selection Results

To determine the best way to integrate context into VLU tasks, we first perform extensive ablation experiments. We analyze various components of our context selection method, including modalities, window size, number of selected frames or scripts, and selection strategies. Finally, we apply our method to benchmark approaches.

**Table 1: (a) Ablation of context and selection modality on Visual SWAG. (b) Ablation of context selection methods. The top half selects context using language models. The bottom half tests the effect of contexts based on the index.**

**(a) Context and Selection Modality**

| Context | Selection | Acc. |
|---|---|---|
| No context | N/A | 74.20 |
| Text | Text | **78.01** |
| Text | Image | 77.87 |
| Text | Text+Image | 77.78 |
| Image | Text | 75.28 |
| Image | Image | 74.97 |
| Image+Text | Text | 77.57 |
| Image+Text | Image | 76.43 |

**(b) Selection Strategy**

| Method | V. SWAG | VCR |
|---|---|---|
| Embedding Sim. | 76.91 | 72.44 |
| Prob Selection | **78.4** | **73.78** |
| Random | 76.79 | 71.43 |
| Index -1 | 77.86 | 71.76 |
| Index -2 | 77.25 | 71.01 |
| Index -3 | 76.15 | 70.23 |

### 6.2.1 Data Modality Ablation.
To ensure the best context utilization, we examine which modalities are most effective for both selecting relevant context and integrating it into VLMs. Table 1a shows results from experiments over VL-BERT with different input and output modalities for our context selection module. The

[1] Please refer to the Supplementary Materials for the detailed implementation steps.

**Table 2: Experiment results of VLMs on Visual SWAG, VCR, and VisualCOMET with/without context. Models with Prob Selection (Prob.) show significant improvement over the baselines. VL-BERT cannot be trained for generative tasks, so results on VisualCOMET are not shown.**

| Model | V.SWAG Acc. | VCR Acc. | VisualCOMET | | |
|---|---|---|---|---|---|
| | | | BLEU4 | CIDER | METEOR |
| VL-BERT | 74.20 | 70.18 | - | - | - |
| VL-BERT+Prob. | **78.40** | **73.78** | - | - | - |
| BLIP | 62.65 | 69.03 | 0.1098 | 0.4468 | 0.1656 |
| BLIP+Prob. | **63.22** | **70.74** | **0.1147** | **0.4595** | **0.1674** |
| BLIP2 | 81.30 | 78.20 | 0.1120 | 0.4492 | 0.1648 |
| BLIP2+Prob. | **84.36** | **81.32** | **0.1163** | **0.4612** | **0.1672** |
| OFA | 54.07 | 69.35 | 0.1329 | 0.4446 | 0.1527 |
| OFA+Prob. | **59.44** | **73.20** | **0.1354** | **0.4642** | **0.1558** |
| MiniGPT-4 | 80.82 | 76.66 | 0.1189 | 0.4503 | 0.1653 |
| MiniGPT-4+Prob. | **83.88** | **79.28** | **0.1217** | **0.4634** | **0.1679** |

**Table 3: Performance Analysis of CARA on CASE. This table shows CARA's detection accuracy on the CASE Set for pseudo-labeled data from VCR and Visual SWAG, highlighting the model's effectiveness and generalizability in detecting samples with insufficient context.**

| Method | Pseudolabelled Data Source | CASE-VCR | CASE-V.SWAG |
|---|---|---|---|
| CARA | VCR | **75.69** | 64.55 |
| | V.SWAG | 54.09 | **73.05** |
| Selector-MaxProb | - | 51.03 | 50.1 |
| Selector-MLP | - | 54.82 | 53.84 |

**Table 4: Experiment results of CARA. We report the improved performance via applying CARA over VLMs across benchmarks. * indicates the performance obtained via applying CARA trained on VCR.**

| Model | V.SWAG Acc. | VCR Acc. | VisualCOMET | | |
|---|---|---|---|---|---|
| | | | BLEU4 | CIDER | METEOR |
| VL-BERT | 73.72 | 70.13 | - | - | - |
| VL-BERT+CARA | **77.04 (74.76*)** | **73.40** | - | - | - |
| BLIP2 | 81.30 | 78.20 | 0.1120 | 0.4492 | 0.1648 |
| BLIP2+CARA | **82.93** | **79.77** | **0.1179** | **0.4642** | **0.1674** |

**Table 5: Generalization of CARA. We apply our abstention detector on base models across various VL benchmarks.**

| Model | VQA v2 | +CARA | GQA | +CARA | OKVQA | +CARA | A-OKVQA | +CARA |
|---|---|---|---|---|---|---|---|---|
| *Zero-shot* | | | | | | | | |
| BLIP2 | 62.5 | **64.9** | 46.33 | **47.57** | 34.68 | **36.55** | 43.94 | **45.00** |
| PNP | 57.52 | **60.42** | 35.68 | **36.96** | 26.98 | **28.54** | 27.78 | **28.43** |
| Prophet | - | - | - | - | 61.1 | **62.28** | 58.20 | **58.37** |
| PromptCap | - | - | - | - | 60.44 | **61.54** | 60.43 | **60.59** |
| MCAN | - | - | - | - | 53.05 | **53.63** | 51.97 | **52.09** |

**Table 6: Verification of Abstained Samples by Human Review. Values are expressed as percentages. "Abstained" refers to the total percentage of validation samples from which the model withheld a prediction. "Ambiguous" and "Insufficient" indicate the percentages of these abstained samples verified by humans as ambiguous or lacking sufficient context, respectively. Note that samples identified as lacking sufficient context are also considered ambiguous, but not vice versa.**

| | Abstention | VQA v2 | GQA | OKVQA | A-OKVQA |
|---|---|---|---|---|---|
| Abstained | | 10.90 | 5.78 | 28.77 | 5.12 |
| Ambiguous | CARA | 69.00 | 70.00 | 64.00 | 69.00 |
| Insufficient Context | | 47.00 | 42.00 | 46.00 | 53.00 |
| Abstained | | 21.07 | 17.05 | 34.08 | 25.08 |
| Ambiguous | Selector-MLP | 23.00 | 16.00 | 25.00 | 17.00 |
| Insufficient Context | | 18.00 | 14.00 | 20.00 | 16.00 |

"Selection modality" column explores the modality used to select context and "Context Modality" refers to the modality of selected context [2]. We find that using text as both the selection and context modality is the most effective approach. This trend holds across different context modalities (image, text, or both).

Regardless of the selection modality, we find adding visual context typically leads to a performance drop. Using text alone for context consistently yields the best results. While the visual context may offer rich information, our findings suggest it often introduces noise, which hurts performance. This highlights an opportunity for future research in integrating multimodal context in VLU tasks. However, one critical finding is that our approach never decreases the performance of the VLM, even in the absence of text for selection or context. To perform this ablation, we use a window size of 3, select one context unit, and use our probabilistic selection method.

*6.2.2 Window Size Ablation.* We next experiment with different context window sizes for VL-BERT on the Visual SWAG and VCR datasets. In this experiment, we limit the number of selected context units to 1, and the window size can range from 0 (no context) to 7. Figure 5a shows how VL-BERT's performance varies with different window sizes. Our results indicate that models with nonzero window sizes outperform the baseline (window size of 0). However, performance plateaus and eventually decreases with excessively large window sizes. The peak accuracies on both VCR and Visual SWAG datasets suggest that their optimal window sizes are 3.

---

[2]Please refer to the Supplementary Materials for details.

*6.2.3 Selection Number Ablation.* Next, we analyze the impact of the amount of context on model performance. Figure 5b presents the results of training VL-BERT, BLIP2, and MiniGPT-4 on VCR and Visual SWAG datasets with different numbers of selected contexts over a window size of 3. In this setup, $M_c$ considers all possible combinations of concatenating $r$ context from $n$ available options arranged temporally rather than being limited to a single optimal context. The models achieved their best performance across all three datasets with a selection number of 2. Notice the drop at the right end of each graph, where the selection number equals the window size. This extreme scenario inputs all the contexts inside the window without a context selection module and shows the importance of our selection module for improving context utilization.

*6.2.4 Selection Strategy Ablation.* In Table 1b, we compare context selection strategies with VL-BERT to determine the most effective one. The bottom of the table presents results from heuristic methods based on context indices, while the top part explores dynamic selection strategies leveraging the embedding similarity. More specifically, we can rely on sentence similarity between the question and text context using Sentence-BERT [34] after determining textual modality as the optimal selection modality. Both the embedding similarity method and heuristic methods are notably outperformed by our jointly trained model, the probabilistic context selection method. In this comparison, selection methods are limited to a window size of 3 and 2 selected contexts.

*6.2.5 Benchmark Comparison.* We apply our probabilistic context selection approach to various base models and report the results in Table 2. With our probabilistic selection method (+ Prob.), all five base models can generally improve their performance across three tasks. Furthermore, the base models can achieve SOTA scores on VisualCOMET with our selection method. These results verify the benefits of incorporating contextual information into VLU tasks and the effectiveness of our method.

## 6.3 Abstention Detector Results

In this section, we discuss the effect of our abstention detector by comparing the performance of VLMs with and without CARA.

*6.3.1 Evaluation of Detection Accuracy for Samples with Insufficient Context.* Building on the confidence-driven pseudo-labeling method outlined in Section 5.2.1, we assembled a small data pool of 500 positive and 500 negative image-question pairs from the VCR validation set and a similar one from Visual SWAG. These datasets were evaluated by Amazon Mechanical Turk workers to ascertain their ambiguity [2]. With this curated data, we created the Context Ambiguity and Sufficiency Evaluation (CASE) Set, spanning both benchmarks to evaluate the efficacy of abstention methods in detecting samples with insufficient context.

We compare CARA to two established methods [50]: Selector-MaxProb, which abstains based on a predefined confidence threshold, and Selector-MLP, which predicts the likelihood of correct predictions using a Multilayer Perceptron module from an image and question. As demonstrated in Table 3, CARA exhibits high detection superior accuracy across these evaluation sets. Notably, when

**Table 7: Abstention evaluation metrics on VQA v2. We show the system risk $\mathcal{R}$, effective reliability $\Phi_1$, and coverage $C$ of three methods. The arrows following the metrics indicate the direction of improvement (for example ($\downarrow$) indicates lower the better). Risks in red are higher than tolerance, and metrics are highlighted in blue when CARA outperforms the baseline methods.**

| VLM | Method | Risk Tolerance $r = 10\%$ | | | Risk Tolerance $r = 30\%$ | | | Risk Tolerance $r = 50\%$ | | |
|---|---|---|---|---|---|---|---|---|---|---|
| | | $\mathcal{R}(\downarrow)$ | $\Phi_1(\uparrow)$ | $C(\uparrow)$ | $\mathcal{R}(\downarrow)$ | $\Phi_1(\uparrow)$ | $C(\uparrow)$ | $\mathcal{R}(\downarrow)$ | $\Phi_1(\uparrow)$ | $C(\uparrow)$ |
| BLIP2 | Selector-MaxProb | 1.4 | 5.0 | 5.2 | 4.9 | 18.7 | 21.2 | 8.5 | 29.3 | 36.8 |
| | Selector-MLP | 2.3 | 7.9 | 8.7 | 7.4 | 25.1 | 30.2 | 11.7 | 36.0 | 46.2 |
| | CARA | 2.6 | 10.0 | 10.7 | 8.8 | 31.1 | 38.8 | 13.1 | 39.0 | 56.3 |
| PNP | Selector-MaxProb | 29.5 | 2.3 | 7.3 | 35.5 | 12.1 | 45.8 | 41.6 | 23.2 | 79.5 |
| | Selector-MLP | 29.0 | 9.8 | 11.6 | 35.4 | 23.5 | 48.9 | 41.4 | 29.8 | 78.5 |
| | CARA | 28.9 | 13.6 | 15.4 | 36.6 | 28.6 | 51.3 | 41.8 | 33.5 | 80.1 |

trained with the pseudo-labeled data from VCR, CARA also demonstrates strong performance on the CASE set of Visual SWAG, underscoring its generalizability. Moreover, the gap between CARA's performance and human judgment (oracle accuracy) underscores the ongoing challenges in detecting samples with insufficient context and highlights the value of the CASE set for future research.

*6.3.2 Performance Enhancement with CARA.* We compare the performance of baseline VLMs with and without CARA across three VLU tasks in Table 4. When using CARA (+CARA), we calculate the accuracy of the baseline VLM only in instances where CARA indicates sufficient context for accurate prediction. The results show that CARA results in a significant improvement in performance across all three tasks. Surprisingly, adding CARA can approach or even exceed the benchmarks set by context-aware models (referenced in Table 2), showing that CARA adds substantial value for multimodal abstention.

*6.3.3 Generalization Across VLU Benchmarks.* We tested CARA's generalizability across four additional VLU benchmarks: VQA v2, GQA, OKVQA, and A-OKVQA. Using the CARA trained on VCR for assessment, we found that CARA significantly improved the performance of baseline VLMs across various tasks and models, as shown in Table 5.

If we further investigate the data points filtered out by CARA and examine them by humans[3], as in table 6, we can observe that the majority of data points filtered out by CARA consist of ambiguous questions and most of them lack sufficient context for a determined answer. These results demonstrate CARA's effectiveness in enhancing model performance across benchmarks and underscore the key problem of instances lacking sufficient context within these benchmarks [4].

*6.3.4 Benchmarking Risk and Coverage.* Previous abstention strategies or selective prediction systems [7, 9] were designed to optimize the balance between risk and coverage, where risk refers to the error rate for the predictions made, and coverage quantifies the total number of predictions issued. An optimal abstention strategy aims to minimize risk while maximizing coverage to the greatest extent possible. We assess the risk ($\mathcal{R}$) and coverage ($C$) performance metrics of CARA compared to previous abstention strategies. Our goal is to minimize risk while maximizing coverage. We also evaluate the

effective reliability ($\Phi_c$) of CARA, rewarding accurate predictions and penalizing incorrect responses.

Table 7 presents the evaluation results with varying risk tolerance levels (i.e. how much risk a model accepts before abstention). We observe that in most cases CARA's $\mathcal{R}$'s system risk is roughly on par with existing methods while achieving significantly higher reliability and effective coverage for both BLIP2 [20] and PNP [43].

# 7 QUALITATIVE EXAMPLES

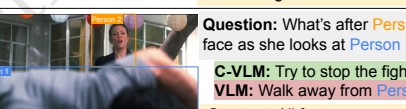

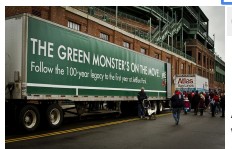

**Figure 6: Qualitative Examples. Correct answers are highlighted in green. Incorrect answers are highlighted in red.**

Figure 6 shows qualitative examples of effective context in VLU (top) and context-aware abstention (bottom).

# 8 CONCLUSION

In this paper, we discussed the issue of insufficient context grappling existing VLU benchmarks and proposed strategies to effectively integrate context, when available, or abstain from speculative prediction in case of samples with insufficient context. We also contributed datasets to enable further exploration of this problem.

# 9 LIMITATION

Please refer to the LIMITATION section in the Supplementary Materials for a detailed discussion.

---

[3]Please refer to the Supplementary Materials for the detailed implementation steps.
[4]The verification results of abstained samples by human review over VCR, Visual-COMET, and Visual SWAG can be found in the Supplementary Materials.

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
