# OpenReview forum: "Detecting Multimodal Situations with Insufficient Context and Abstaining from Baseless Predictions"
_acmmm.org/ACMMM/2024/Conference — MM2024 Poster_

### Official Review · Reviewer_mMFQ · 2024-05-23

**Rating:** 5
**Confidence:** 2

**Summary:**

This work identifies a pervasive issue of insufficient event-specific context in common Vision Language Understanding benchmarks. To address this issue, authors introduce a novel context selection method. In addition, authors develop a method for abstaining on samples lacking necessary context. About the data contribution, authors collect contextual data for VCR, SWAG, and VisualCOMET, and create a context ambiguity and sufficiency evaluation set annotated with the labels (sufficient or insufficient context) for insufficient context detection.

**Strengths:**

This work identifies a prevalent issue in VLU benchmarks, and proposes the corresponding data-centric methods to incorporate relevant context and abstain on samples lacking context. In addition, authors make data contribution, including collecting contextual data for context-aware model prediction and creating a context ambiguity and sufficiency evaluation set for insufficient context detection.

**Limitations:**

1.	Authors are encouraged to provide brief descriptions about evaluation metrics including the system risk $\mathcal{R}$, effective reliability $\Phi_1$ and coverage $\mathcal{C}$.
2.	Authors are encouraged to perform the abstention evaluation in Table 7 on more VLMs like Pythia, ViLBERT, VisualBERT, CLIP-ViL and OFA, so as to have a fair comparison with results in the work [A] and [B].
3.	This work focuses on the visual question answering task. Can the propose method be applied into other vision language understanding tasks, e.g., visual entailment?

[A] Spencer Whitehead, Suzanne Petryk, Vedaad Shakib, Joseph Gonzalez, Trevor Darrell, Anna Rohrbach, and Marcus Rohrbach. 2022. Reliable visual question answering: Abstain rather than answer incorrectly. In European Conference on Computer Vision. Springer, 148–166.

[B] Corentin Dancette, Spencer Whitehead, Rishabh Maheshwary, Ramakrishna Vedantam, Stefan Scherer, Xinlei Chen, Matthieu Cord, and Marcus Rohrbach. 2023. Improving Selective Visual Question Answering by Learning From Your Peers. In Proceedings of the IEEE/CVF Conference on Computer Vision and Pattern Recognition (CVPR). 24049–24059.

**Suitability:**

3

---

### Official Review · Reviewer_WBMX · 2024-05-24

**Rating:** 4
**Confidence:** 3

**Summary:**

This paper analyses current VLU benchmarks and reveals that some samples has answers that rely on unsupported assumptions provided by the context. It try to address this problem by two approaches: 1) collect contextual data and train a model to help selecting contexts; 2) train model to detect insufficient context samples and improve model performance by response abstention.

**Strengths:**

1. The experiments are extensive which cover different vision-language models and VLU benchmarks.

2. The paper is well-structured and easy to follow.

**Limitations:**

1. Missing an important VLM, LLaVA, in the experiment.

2. Results in Table 3. seem not to be convincing to show the generalizability of detector.

3. It is unkown how to choose the $\mu$ and $\gamma$.

4. In Table 7, CARA do not show a consistently better performance than Selector-MLP in terms of Risk.

**Suitability:**

3

---

### Official Review · Reviewer_vMW1 · 2024-05-25

**Rating:** 5
**Confidence:** 2

**Summary:**

This paper focuses on the detection of multimodal situations with insufficient context and the importance of abstaining from making baseless predictions in VLMs. This work introduces an abstention mechanism in multimodal systems to avoid making predictions when there is insufficient context to answer a question, without the need for re-training. The experimental results show the effectiveness of the proposed Context-AwaRe Abstention (CARA) detector.

**Strengths:**

1. Helping VLMs detect situations where they should not respond can improve the model's reliability. It is valuable for fields with high trust requirements.
2. This work provides extensive related analyses, is well-written and easy to read, and proposes a detection method that does not require retraining.
3. The experimental results show the effectiveness of the proposed method.

**Limitations:**

1. In the paper, CARA only compare MaxProb and MLP. Could the authors provide some more in-depth abstention strategies for comparison to demonstrate the effectiveness of CARA?

**Suitability:**

3

---

### Official Review · Reviewer_rY9e · 2024-05-29

**Rating:** 3
**Confidence:** 3

**Summary:**

The paper introduces a model-agnostic Context-AwaRe Abstention detector detection module, CARA, designed to address the issue of contradictory context and answer hypotheses in visual-text understanding datasets. CARA identifies samples that lack sufficient contextual information and aims to enhance model accuracy by abstaining from making predictions on such samples. Additionally, the authors have developed a benchmark dataset named CASE for evaluating the performance of context detection modules in terms of ambiguity and sufficiency of context.

**Strengths:**

- The motivation is compelling and holds significant value for ensuring that visual-language models produce credible and evidence-based outputs in complex real-world scenarios.
- The work is substantial, presenting an effective and feasible context detection module and a benchmark dataset.
- The writing is fluent, and the methodology is relatively clear.

**Limitations:**

1. The exploration of visual context within the proposed method seems limited, with experimental results indicating a predominance of textual context utilization. This reliance on text could lead to a perception that the approach is more aligned with text-centric models like RAG, particularly if CARA does not include a mechanism for assessing the adequacy of contextual information to determine the feasibility of providing an answer. The authors should consider expanding the method's capabilities to more effectively integrate and leverage visual context alongside textual information.
2. The CASE dataset section lacks detail, particularly in terms of dataset-related statistics. For instance, how many context image-text pairs correspond to each image-text pair on average? Based on the numbers provided, it seems that an image-text pair cannot be allocated more than two context image-text pairs on average. Given that this dataset is a significant innovative aspect of the work, a more thorough explanation of its construction reliability is needed. More details in this section would be beneficial.
3. There is some ambiguity in the method section of the paper. Does the training of the Context Selection Module include parameter updates to the underlying Visual Language Model (VLM)? Clarification on this point would be appreciated.

**Suitability:**

2

---

### Meta-Review · Area_Chair_RR6J · 2024-07-04

**Recommendation:** Accept (Poster)
**Confidence:** 4

**Metareview:**

This submission addresses the issue of unsupported assumptions in Vision-Language Understanding (VLU) benchmarks by collecting contextual data and training a context selection module, which leads to improvements across multiple benchmarks. The proposed Context-AwaRe Abstention (CARA) detector enhances model accuracy by abstaining from responding to samples lacking sufficient context and shows generalization to new benchmarks.

Three out of four reviewers tend to accept, while Reviewer rY9e still has concerns about the proposed method. All reviewers acknowledge that the proposed CARA can help visual-language models produce reliable output. Their initial concerns include hyperparameter choice, the need for more VLM models, and the predominance of textual context utilization. The rebuttal partially addresses these concerns. After the rebuttal, two reviewers are willing to maintain their recommendation of acceptance. I agree with Reviewer rY9e regarding the potential bias toward the utilization of textual information and believe the submission should discuss how visual context is utilized and defined. Considering this submission addresses an interesting problem and proposes a simple method, I would recommend acceptance. Please prepare for revision carefully and incorporate the suggested changes.